# The Importance of the Type of Posterior Staphyloma in the Development of Myopic Maculopathy

**DOI:** 10.3390/diagnostics14151581

**Published:** 2024-07-23

**Authors:** Jorge Ruiz-Medrano, Mariluz Puertas, Ignacio Flores-Moreno, Elena Almazán-Alonso, María García-Zamora, Bachar Kudsieh, José M. Ruiz-Moreno

**Affiliations:** 1Puerta de Hierro-Majadahonda University Hospital, C/Manuel de Falla, 1, 28222 Madrid, Spain; mariluzpuertas@gmail.com (M.P.); i_floresmoreno@hotmail.com (I.F.-M.); elena.almazanalonso@gmail.com (E.A.-A.); maria.gciazamora@gmail.com (M.G.-Z.); josemaria.ruiz@uclm.es (J.M.R.-M.); 2IMO Ocular Microsurgery Institute, Miranza Corporation, 28035 Madrid, Spain; 3Clínica Suárez Leoz, 28010 Madrid, Spain; 4Department of Ophthalmology, Castilla La Mancha University, 02001 Albacete, Spain

**Keywords:** myopic maculopathy, posterior staphyloma, ATN, pathologic myopia

## Abstract

The objective of this paper was to determine how different types of posterior staphyloma (PS) may affect the appearance and degree of myopic maculopathy. A cross-sectional study was conducted, in which 467 eyes from 246 highly myopic patients [axial length (AL) ≥ 26 mm] were studied. A complete ophthalmic exploration was carried out on all patients, including imaging tests. The presence of macular PS was established as the main comparison variable between groups (macular PS vs. non-macular PS vs. non-PS). The variables analyzed included age, AL, decimal best-corrected visual acuity (BCVA), Atrophy (A)/Traction (T)/Neovascularization (N) components according to the ATN grading system, and the presence of severe pathologic myopia (PM). Out of the total, 179 eyes (38.3%) presented macular PS, 146 eyes presented non-macular PS (31.2%), and 142 eyes showed no PS (30.4%). The group without PS was significantly younger than macular PS and non-macular PS groups (53.85 vs. 66.57 vs. 65.20 years; *p* < 0.001 each, respectively). There were no age differences between PS groups. Eyes with macular PS (31.47 ± 2.30 mm) were significantly longer than those with non-macular PS (28.68 ± 1.78 mm, *p* < 0.001) and those without PS (27.47 ± 1.34 mm, *p* < 0.001). BCVA was significantly better in the non-PS group (0.75 ± 0.27) compared to the non-macular PS (0.56 ± 0.31) and macular PS groups (0.43 ± 0.33), with *p* < 0.001 each. Eyes without PS showed significantly lower A and T components (1.31 ± 0.96 and 0.30 ± 0.53, respectively) than non-macular PS (2.21 ± 0.75 and 0.71 ± 0.99, respectively, *p* < 0.001 each) and macular PS eyes (2.83 ± 0.64 and 1.11 ± 1.10, respectively, *p* < 0.001 each). The N component was lower in non-PS eyes vs. non-macular PS eyes (0.20 ± 0.59 vs. 0.47 ± 0.83, *p* < 0.001) and as compared to the macular PS group (0.68 ± 0.90, *p* < 0.01). Additionally, the N component was significantly lower in the non-macular PS group than in the macular PS one (*p* < 0.05). The prevalence of severe PM was different between groups (*p* < 0.001). It was higher among macular PS eyes (138/179) when compared to other groups (*p* < 0.001, each), followed by the non-macular PS eyes (40/146) and being the lowest in the non-PS group (20/142). To conclude, macular PS is associated with a more advanced maculopathy, worse vision, and higher rates of severe PM.

## 1. Introduction

It has been recently published by our group that posterior staphyloma (PS) was associated with a more advanced maculopathy, worse vision, and higher rates of severe pathologic myopia (PM) [1]. Additionally, this study also found that axial length (AL), followed by age, constituted the most relevant characteristic associated with the prevalence of PS [1]. These findings have been reported in other studies, confirming that eyes with high myopia and PS have worse visual acuity than highly myopic eyes without PS [2,3,4,5,6].

Eyes showing PS were significantly associated with advanced myopic maculopathy and lower vision [7,8]. Therefore, it is currently accepted that PS is the main determining factor for myopic maculopathy, which remains the most common cause of visual acuity loss in highly myopic patients [1,2,3,7,9,10,11,12,13,14]. AL has usually been used to define and quantify the degree of myopia [15]. However, when evaluating patients with high myopia, it is essential to determine the presence of PS and its subtypes. Curtin [16] proposed a detailed classification system, which has been recently modified by Ohno-Matsui [2].

Primary staphyloma subtypes I and II, as well as all compound subtypes of PS, involve the macular area [2,16]. In such cases, AL represents the longest length of the ocular globe and, therefore, constitutes an excellent biomarker of the size of the eyeball [7]. However, in other staphyloma subtypes (subtypes III, IV, and V), this deformation does not primarily affect the macula, so there is currently no reliable and objective method to determine the longest (non-axial) eye length. Furthermore, these differences would imply variable involvement in the degree of myopic maculopathy; i.e., cases in which PS does not affect the macula should not be expected to develop high degrees of myopic maculopathy, as PS is mainly located outside of the macular region [7]. This issue might be considered a potential bias of previous studies [1,7]. To avoid this limitation and to increase the accuracy of the study of highly myopic patients, it would be ideal to separately analyze, on the one hand, highly myopic patients with PS subtypes I and II and its compounds that, as mentioned above, involve the macular region and where the AL represents a reliable biomarker; on the other hand, it would be ideal to study PS subtypes III, IV, and V, in which the absence of macular alterations induced by the PS’s mean AL is no longer a reliable measure of the globe’s longest axis. This would mean that the specific type of PS, and not only its presence, is what really influences the appearance and degree of myopic maculopathy.

The objective of this paper was to determine how different types of posterior staphyloma (PS) may affect the appearance and degree of myopic maculopathy.

## 2. Materials and Methods

A cross-sectional study was performed following the Tenets of the Declaration of Helsinki. The Ethics Committee of Puerta de Hierro-Majadahonda University Hospital (Madrid, Spain) reviewed and approved the protocol of this study under number PI 43/20. The appropriate informed consent was signed by patients > 18 years old; parents/guardians signed in cases of underage patients. The examination guidelines followed previous publications by our investigation group [1]. Patients with high myopia who attended the outpatient clinic at Puerta de Hierro-Majadahonda University Hospital were examined. Inclusion criteria were a diagnosis of high myopia (AL ≥ 26 mm); clear media; and good quality imaging by software standards.

Exclusion criteria were a dome-shaped macula; previous intraocular surgery (refractive or cataract surgery were allowed); or other ocular or systemic diseases, including uveitis, diabetic retinopathy, punctate inner choroidopathy, glaucoma, multifocal choroiditis, Marfan syndrome, angioid streaks, retinal vein/artery occlusion, and/or an idiopathic macular hole (MH). In total, 2 retina experts separately evaluated all cases. In cases of disparity, the eyes were excluded. Both eyes were independently examined if they met the inclusion criteria for this study. The following three groups were considered: macular PS (Types I, II and VI), non-macular PS (types III, IV and V), and non-PS eyes. Demographic data were obtained from patients’ clinical records. All study participants underwent a complete ophthalmological examination including decimal best-corrected visual acuity (BCVA) assessment under cycloplegia, slit-lamp anterior segment examination, optical biometry (IOL Master-500/700, Carl Zeiss Meditec AG, Jena, Germany), intraocular pressure (Goldman applanation tonometry), and indirect fundus ophthalmoscopy.

### 2.1. Multimodal Imaging

All study subjects underwent multimodal imaging testing that included color fundus photography using Zeiss Clarus TM 500 (Carl Zeiss Meditec AG, Jena, Germany), DRI-OCT Triton^®^ (Topcon Corporation, Tokio, Japan), and/or Optos Optomap Panoramic 200A imaging system (Optos PLC, Dunfermline, UK). Fundus autofluorescence was obtained with Spectralis^®^ OCT (Heidelberg Engineering Inc., Heidelberg, Germany); OCT was carried out with DRI-OCT Triton^®^ plus, and the examination protocol included radial 12 mm scans centered on the fovea containing 1024 axial scans each. Fluorescein angiography and/or OCT-angiography were performed when in doubt in cases of suspected myopic choroidal neovascularization (CNV).

### 2.2. ATN Grading System

The ATN grading system [10,17] for myopic maculopathy was applied in all cases and is detailed in Table 1. Two masked retinal specialists independently assessed atrophic (A), tractional (T), and neovascular (N) components of the disease based on fundus photography and SS-OCT scans. N2a and N2s represent dynamic stages; both were considered N2 for statistical purposes (eyes classified as N2a before treatment can become N2s after treatment; N2s can resume activity in time). A presurgical T score was considered for ATN grading in patients who had previously undergone pars plana vitrectomy due to tractional myopic maculopathy (the impact of surgery on the tractional component was not considered).

Based on the original ATN grading system from *Ruiz-Medrano J. Prog Retin Eye Res*. **2019**, *69*, 80–115 [1].

In addition, eyes with myopic CNV or myopic full-thickness MH were included in this study as long as the atrophic component was equal to or higher than diffuse atrophy maculopathy (A2). This criterion was applied to prevent including those secondary to other pathologies.

### 2.3. Pathologic Myopia and Severe Pathologic Myopia

All the eyes were classified as PM or severe PM based on ATN grading criteria as described in previous studies [17]. To summarize, PM was defined as the presence of an atrophic component ≥ A2, and severe PM was defined as myopic maculopathy ≥A3, ≥T3, and/or N2 (active myopic CNV, scar myopic CNV, or Fuchs spot).

### 2.4. Posterior Staphyloma

The presence of PS and its subtype was determined using the combination of indirect ophthalmoscopy, fundus photography, and OCT radial b-scan (Figure 1, Figure 2 and Figure 3). Two retinal specialists independently determined the presence/absence and, if present, the subtype of PS. The variables of age, AL, BCVA, myopic maculopathy according to ATN classification, and the presence of severe PM were compared among the three study groups (i.e., macular PS; non-macular PS; and non-PS). 

### 2.5. Statistical Analysis

All analyses were performed using the IBM-SPSS statistical software (IBM-SPSS, v. 28.0.0.0, Chicago, IL, USA). A two-tailed *p* value < 0.05 was considered statistically significant. Descriptive statistics were provided for normally distributed variables as means, with standard deviation (SD) for quantitative variables and n (percentage) for categorical variables. Variables were tested for normal distribution using the Kolmogorov–Smirnov test.

Demographic data, age, BCVA, AL, ATN grading, and severe PM were compared between groups. Categorical variables were compared using the Chi-square test for normally distributed variables and Fisher’s exact test for non-parametric variables. Normally distributed continuous variables were compared using the independent samples Student’s T-test when two groups were studied and using the analysis of variance (ANOVA) test if more than two groups were compared. The Kruskal–Wallis test was used to compare the following ordinal variables between groups: A, T, and N components.

Interobserver agreement for the ATN grading system and PS classification were analyzed. As these variables were categorical, a concordance correlation coefficient was obtained; the rho coefficient was considered moderate (≥0.4), good (≥0.6), or excellent (≥0.8).

## 3. Results

This study included 467 highly myopic eyes of 246 patients; 69.2% of them were women. The mean age was 62.27 ± 13.53 years (range: 14–92 years), mean AL was 29.38 ± 2.54 mm (range: 26.00–37.60 mm), and mean BCVA was 0.57 ± 0.33 decimal (range: 0.001–1.0 decimal). Regarding myopic maculopathy and according to the ATN classification, the enrolled eyes showed a mean A component of 2.28 ± 1.0 (range: 0–4), T component of 0.73 ± 0.98 (range; 0–5), and N component of 0.47 ± 0.82 (range: 0–2). PM and severe PM were found in 168 (35.97%) and 214 eyes (45.82%), respectively. Demographic and clinical characteristics are shown in Table 2.

The interobserver concordance correlation for PS subtype and ATN classification was excellent; the rho coefficients obtained were as follows: PS subtype: 0.98 (*p* < 0.001), A component: 0.98 (*p* < 0.001), T component: 0.98 (*p* < 0.001), and N: 0.99 (*p* < 0.001).

Out of the total, 179 (38.3%) eyes presented macular PS, 146 (31.2%) eyes presented non-macular PS, and 142 (30.4%) eyes did not show PS (Table 3). The eyes without PS were younger than the eyes in the macular and non-macular PS groups (53.85 ± 14.01 versus (vs). 66.57 ± 12.07 vs. 65.20 ± 10.83 years old; *p* < 0.001 each). Nonetheless, the age between the macular PS vs. non-macular PS groups was not significantly different (*p* = 0.95, Table 3).

Regarding AL, the longest eyes were those with macular PS (31.47 ± 2.30 mm), followed by non-macular PS eyes (28.68 ± 1.78 mm) and by those without PS (27.47 ± 1.34 mm). Statistically significant differences were found among all three groups (*p* < 0.001, Table 3) and when they were compared pairwise (*p* < 0.001, each) (Table 3).

The BCVA was significantly better among eyes without PS (0.75 ± 0.27 decimal) compared to non-macular PS (0.56 ± 0.31 decimal) and also to macular PS, which showed the worst BCVA results (0.43 ± 0.33 decimal). Differences between all groups and differences when they were compared pairwise were statistically significant (*p* < 0.001, Table 3).

Regarding myopic maculopathy according to ATN grading, macular PS eyes showed higher scores for both A and T components (2.83 ± 0.64 and 1.11 ± 1.10, respectively), compared to non-macular PS eyes (2.21 ± 0.75 and 0.71 ± 0.99, respectively, *p* < 0.001 each) and to non-PS groups (1.31 ± 0.96 and 0.30 ± 0.53, respectively, *p* < 0.001 each), which showed the lowest scores (Table 3).

The N score was significantly lower in the eyes of the group with no PS (0.20 ± 0.59) than in non-macular PS eyes (0.47 ± 0.83), while macular PS eyes had the greatest N score (0.68 ± 0.90). Differences among groups were statistically significant (*p* < 0.001, Table 3) as well as pairwise comparisons (Table 3). Significant differences between non-PS eyes and both macular PS (*p* < 0.001) and non-macular PS (*p* < 0.01) eyes, as well as between macular and non-macular PS groups (*p* < 0.05), were observed (Table 3).

The prevalence of severe PM was statistically significantly different among the three groups (*p* < 0.001, Table 3). The prevalence was higher in the macular PS eyes (138/179) than in the non-macular PS eyes (40/146) and in those without PS (20/142) (*p* < 0.001 each, respectively); whereas there were no significant differences between non-macular PS eyes and those without PS (Table 3).

## 4. Discussion

Current evidence, including our previous studies, has demonstrated the relevance of PS in the development of myopic maculopathy [1,2,3,4,5,6,7,8,9,10,11,12,13,14]. PS was first described by Antonio Scarpa in 1801 in two post-mortem eyes, finding that the posterior portion of the eyeballs showed significant prominence. In 1977, Curtin [16] classified PS in pathologically myopic eyes into 10 different types. Types I to V were considered primary staphylomas, and types VI to X were termed compound staphylomas [16]. This is still the most commonly used classification to describe the fundus of the high myope in daily clinical practice. However, this classification based on ophthalmoscopic imaging alone is not always easy to perform accurately and has a high component of subjectivity; moreover, images obtained with OCT in these highly myopic eyes with PS do not always agree with Curtin’s classification [16]. Ohno-Matsui made an objective classification of PS using different imaging modalities [2]. In order to study PS color fundus photography [14], ultrasonography [4] and magnetic resonance imaging can potentially be used and provide very useful results according to the authors [2].

Fundus imaging using Optos widefield technology is considered a novel and highly effective method of visualizing the full extent of staphyloma, which cannot be assessed with conventional 50° fundus photography, and although the use of MRI may be more accurate, the information obtained by Optos would be more usable in clinical settings. Ohno-Matsui showed that approximately half of the eyes with high myopia had no PS on 3D MRI. In these eyes without staphyloma, the AL was simply elongated in the anteroposterior direction, with no deformation (PS) forming in the posterior portion of the globe. These eyes were barrel-shaped, L-shaped, and would match our group with LA greater than 26 mm but without PS, which, in our case, was 30.41% (142 eyes) [1]. It is difficult to compare the incidence of PS between studies due to differences in the definition of PS, the methods used to detect it, and the characteristics of the patients used. Curtin and Karlin [6] reported a 1.4% incidence of PS in eyes with an AL of 26.5 to 27.4 mm and up to 71.4% in eyes with an AL of 33.5 to 36.6 mm. Chang et al. examined 359 adults aged 40 years with myopia greater than −6D using 45° retinography, detecting PS in 11.1% of eyes with −6 to −7.99D, 23.8% of eyes with 8 to 9.99D, and 59.0% of eyes with more than −10.0D. In our case, our study included 467 highly myopic eyes of 246 patients, with a mean AL of 29.38 ± 2.54 mm; among them, 325 eyes (69.59%) had PS [1]. Curtin reported that AL varied significantly even in eyes with the same type of staphyloma and suggested that AL might not be a good indicator to characterize pathological myopia. Foveal AL represents an accurate and reliable method for assessing the ocular globe length in eyes with macular PS subtypes I and II and most compound types. However, AL does not constitute a perfect biomarker of the ocular globe length in those PS subtypes that do not affect the macula region (subtypes III, IV, and V), and, therefore, its use may have led to biased results in previous studies where all subtypes of PS have been analyzed together [1,7].

In order to prevent this bias, this study separately evaluated eyes with staphylomas that involved the macula vs. other subtypes. Hence, the results of this study could provide more reliable information regarding the relevance of PS and its relationship with AL and the development of myopic maculopathy. Comparing the results of this current study with those of our group’s previous research [1], and while using the same control group, the differences in AL were much more pronounced when macular PS eyes were compared to those without PS (31.47 vs. 27.47 mm) and were more subtle between other types of PS and non-PS eyes (28.68 vs. 27.47 mm). Previous data, which included all types of PS in the analysis, showed significantly different AL values (30.21 vs. 27.47 mm) [1]. Calculations of the longest measurement of the eye is accurate in eyes with macular PS and without PS, but the axial length in those eyes with PS subtypes III, IV, and V may not represent the longest length of the eye, as the deepest part of the staphyloma is not centered on the macula. Therefore, AL is not an accurate biomarker in such cases, for which other tests should probably be proposed to determine the real globe length, such as magnetic resonance imaging.

Magnetic resonance imaging was previously used by Ohno-Matsui et al. in 105 patients with pathologic myopia. The eyes were examined by 3D MRI and Optos, allowing for the conclusion that 3D magnetic resonance imaging was useful in analyzing the shapes of eyes with and without staphyloma, and if we correlated the MRI data to Optos images, we can obtain useful information on the relationship between the eye shape and the fundus appearances.

Despite all of the above, the age of the macular PS group (66.57 years) was comparable to those of the overall PS group (65.95 years) [1] and to the non-macular PS group (65.20 years). As expected, based on previous studies’ results, patients whose eyes were without PS were younger (53.85 years, Table 3). This is important, as previous reports suggest that age is only second to axial length as a risk factor for the onset of PS. PS is extremely rare in infants, and its prevalence grows with age, as does the degree of myopia of these patients [1]. The appearance and progression of PS, in time, is what seems to lead to atrophic, tractional, and neovascular changes.

Similarly, when comparing data of the A, T, and N components of myopic maculopathy, the eyes with macular PS showed a greater degree of myopic maculopathy according to all three parameters analyzed than when considering the group with all types of PS combined (ATN components: 2.83 vs. 2.55, 1.11 vs. 0.93, and 0.68 vs. 0.59; respectively) and as compared to non-macular PS group (ATN components: 2.21, 0.71 and 0.47; respectively). The differences between the PS and non-PS groups increased significantly when evaluating only those PS that affected the macular region instead of the group with all the PS subtypes combined [1], and also vs. non-macular PS. Macular PS eyes showed higher A and T scores (2.83 and 1.11, respectively) than non-macular PS eyes (2.21 and 0.71, respectively), whose scores were higher than in those of patients without PS (1.31 and 0.30, respectively). Differences among all three groups and pairwise comparisons were statistically significant (*p* < 0.001, Table 3).

Eyes with PS showed higher N scores as well, finding differences between both PS groups and making myopic choroidal neovascularization more common among eyes with macular PS. BCVA was slightly greater in the global PS group than in the macular PS group (0.49 vs. 0.43 decimal) [1]. Macular PS eyes showed the worst BCVA of all groups, with significant differences compared to non-macular PS and non-PS groups (0.43 vs. 0.56 vs. 0.75 decimal, respectively; Table 3). As expected, the presence of macular PS resulted in a higher degree of myopic maculopathy and, thus, worse BCVA. These data matched previous studies’ results, confirming worse BCVA data among eyes with PS [1,7]. Additionally, eyes with macular PS showed a higher risk of being classified as severe PM [17] vs. the two other groups (Table 3).

These data agreed with Ohno-Matsui’s [2], confirming macular PS as the greatest risk factor for visual loss in eyes with PM. In those cases of PS subtypes III, IV, and V, eyeball length measurement using an accurate and reliable biomarker remains a pending issue. Understanding the maximum length of the ocular globe is important, especially in patients with PS without macular involvement, as the lower incidence and severity of myopic maculopathy in these eyes reduces its clinical relevance. So, although it would be ideal for research purposes, the measurement of foveal axial length should still be useful in daily clinical practice.

The main limitation of this study was that the determination of the presence of PS could not be assessed using imaging techniques such as widefield OCT or a widefield fundus camera as in previous papers [1,7]. The presence of such a feature in this study was established by two retina experts combining indirect ophthalmoscopy with the findings of SS-OCT and wide fundus photography in certain cases. The second limitation was the already mentioned low reliability of AL calculation in eyes showing PS subtypes III, IV, and V without macular involvement, using foveal AL.

In conclusion, macular PS was specifically identified as the greatest risk factor for visual loss and was associated with a higher degree of myopic maculopathy, worse visual acuity, and a greater prevalence of severe PM. These differences were more pronounced when evaluating eyes with macular PS vs. non-macular PS eyes.

## Figures and Tables

**Figure 1 diagnostics-14-01581-f001:**
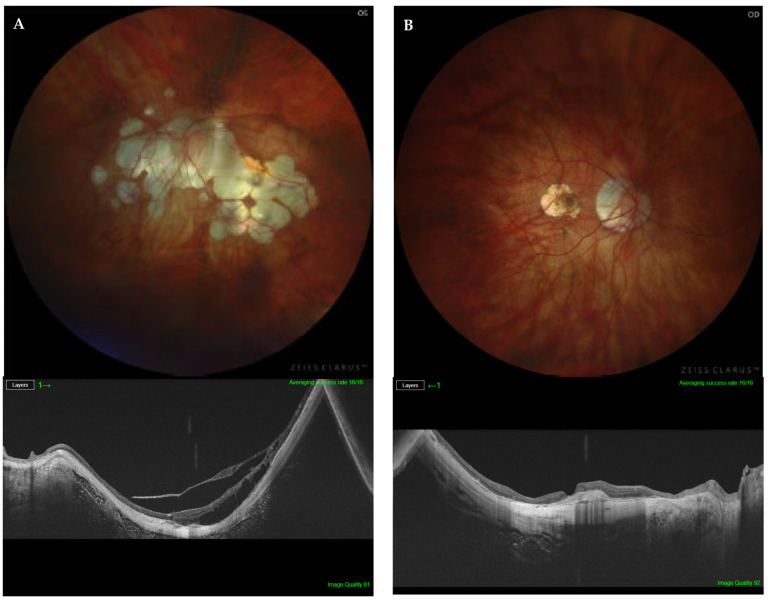
(**A**) Fundus photography of the left highly myopic eye of a 75-year-old patient, with macular posterior staphyloma (PS) type 9, AL of 32.16 mm, and BCVA of 0.3 (top). Structural optical coherence tomography showing important schisis and traction (bottom). (**B**) Fundus photography of the right highly myopic eye of a 67-year-old patient, with macular PS type 2, AL of 30.31 mm, and BCVA of 0.2 (top). Structural optical coherence tomography showing a CNV scar and adjacent atrophy (bottom).

**Figure 2 diagnostics-14-01581-f002:**
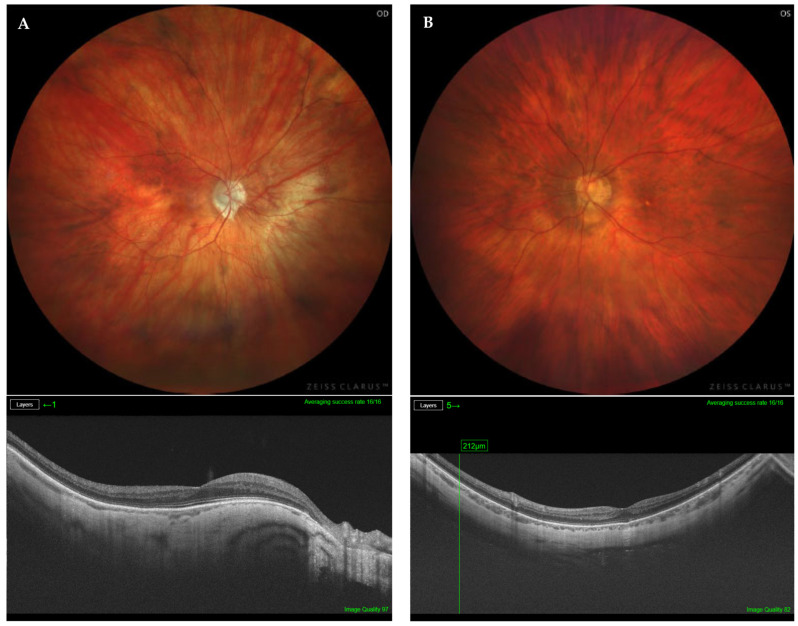
(**A**) Fundus photography of the right highly myopic eye of a 70-year-old patient, with type 5 non-macular PS, AL of 29.21 mm, and BCVA of 0.7 (top). Structural optical coherence tomography showing a normal retinal profile (bottom). (**B**) Left highly myopic eye of a 62-year-old patient, with type 3 non-macular PS, AL of 29.75 mm, and BCVA of 0.8 (top). Structural optical coherence tomography showing a normal retinal profile (bottom).

**Figure 3 diagnostics-14-01581-f003:**
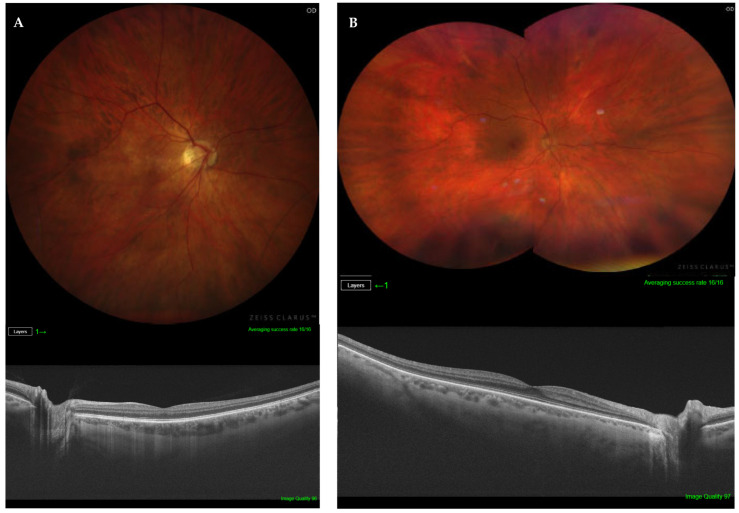
(**A**) Right eye with high myopia and non-PS (top). Structural optical coherence tomography showing a normal retinal profile (bottom). (**B**) Right eye montage of a non-PS highly myopic eye in a 64-year-old patient, with AL of 26.27 mm and BCVA of 0.9 (top). Structural optical coherence tomography showing a normal retinal profile (bottom).

**Table 1 diagnostics-14-01581-t001:** Updated ATN classification system.

Atrophic Component (A)	Tractional Component (T)	Neovascular Component (N)
A0: no myopic retinal lesions	T0: No macular schisis	N0: No myopic CNV
A1: tessellated fundus only	T1: Inner or outer foveoschisis	N1: Lacquer cracks
A2: diffuse chorioretinal atrophy	T2: Inner + outer foveoschisis OR lamellar macular hole	N2a: Active CNV
A3: patchy chorioretinal atrophy	T3: Foveal detachment	N2s: Scar/Fuch’s spot
A4: complete macular atrophy	T4: Full-thickness MH	
	T5: MH + Retinal detachment	

CNV: choroidal neovascularization; MH: macular hole.

**Table 2 diagnostics-14-01581-t002:** Demographic and clinical characteristics of the overall study population included in this study.

Variable	n	Mean	Standard Deviation	Minimum	Maximum
**Age**	467	62.27	13.53	14	92
**AL**	458	29.38	2.54	26	37.60
**BCVA**	467	0.57	0.33	0.001	1.0
**A**	467	2.28	1.00	0	4
**T**	467	0.73	0.98	0	5
**N**	467	0.47	0.82	0	2

AL: axial length; BCVA: best-corrected visual acuity; and A, T, and N: atrophic, traction, and neovascular components of the ATN classification of myopic maculopathy.

**Table 3 diagnostics-14-01581-t003:** Comparison of highly myopic eyes without posterior staphyloma; macular posterior staphyloma; and non-macular posterior staphyloma.

Variable	Posterior Staphyloma	N	Mean	Standard Deviation	*p* *	Pairwise Comparisons
**Age**	No	142	53.85	14.01	<0.001 *	<0.001 *
	Macular	179	66.57	12.07		<0.001 *
	Non-macular	146	65.20	10.83		0.95 *
**AL**	No	140	27.47	1.34	<0.001 *	<0.001 *
	Macular	175	31.47	2.30		<0.001 *
	Non-macular	143	28.68	1.78		<0.001 *
**BCVA**	No	142	0.75	0.27	<0.001 *	<0.001 *
	Macular	179	0.43	0.33		<0.001 *
	Non-macular	146	0.56	0.31		0.001 *
**A**	No	142	1.31	0.96	<0.001 **	<0.001 **
	Macular	179	2.83	0.64		<0.001 **
	Non-macular	146	2.21	0.75		<0.001 **
**T**	No	142	0.30	0.53	<0.001 **	<0.001 **
	Macular	179	1.11	1.10		<0.001 **
	Non-macular	146	0.71	0.99		<0.001 **
**N**	No	142	0.20	0.59	<0.001 **	<0.001 **
	Macular	179	0.68	0.90		<0.01 **
	Non-macular	146	0.47	0.83		<0.05 **
**SPM**	No	20/142			<0.001 ***	<0.001 ***
	Macular	138/179				>0.05 ***
	Non-macular	40/146				<0.001 ***

AL: axial length; BCVA: best-corrected visual acuity; A, T, and N: atrophic, traction, and neovascular components of ATN classification of myopic maculopathy; SPM: severe pathologic myopia. * ANOVA test, ** Kruskal–Wallis test, *** Chi-square test.

## Data Availability

Data for this study will be available upon request.

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
