# Peer review of "The Importance of the Type of Posterior Staphyloma in the Development of Myopic Maculopathy"

_diagnostics, 2024, doi:10.3390/diagnostics14151581_

Round 1

Reviewer 1 Report

Comments and Suggestions for Authors

Interesting paper. If possible, I would add the values of Intraocular pressure, as one possible cause of myopia with no posterior staphyloma is an elevated IOP in te developmental age.

In addition, can you comment about the worsening of myopia with age, and the possible development of atrophic zones with age?

Author Response

Interesting paper. If possible, I would add the values of Intraocular pressure, as one possible cause of myopia with no posterior staphyloma is an elevated IOP in te developmental age.

Thank you very much for your accurate comment. Unfortunately, we do not have previous records and any patient with uncontrolled IOP was excluded from the study. We agree this is something that could indeed improve the consistency of the paper and will work on it for future studies

In addition, can you comment about the worsening of myopia with age, and the possible development of atrophic zones with age?

As suggested by the reviewer comments on the suggested topic were added to the discussion in page 14, paragraph 3 (..."This is important, as previous reports suggest that age is only second to axial length as a risk factor for the onset of PS. PS is extremely rare in infants and its prevalence grows with age as does the degree of myopia of these patients.1 The appearance and progression of PS, in time, is what seems to lead to atrophic, tractional and neovascular changes."). 

Reviewer 2 Report

Comments and Suggestions for Authors

I agree the content of the manuscript itself. However, in introduction, results, and discussion sections, each sentences are not well organized in each paragraphs. Please check and make up more clearly each paragraphs.

Minor revision   1.Although this study mentions that all subjects undergo multimodal imaging, the image shown here are only color fundus images. Figures should be re constituted displayed by multimodal image data for each patients. Using photoshop is strongly recommended. 2. Detailed explanation of ATN grading is required in method section. 3. P value of Table 2 seems to have less meaning. Authors should consider incorporating Table 3 into Table2.

Author Response

I agree the content of the manuscript itself. However, in introduction, results, and discussion sections, each sentences are not well organized in each paragraphs. Please check and make up more clearly each paragraphs.

We would like to thank the reviewer for this comment. As the same was requested by another reviewer, sections were reorganized and reduced in number across all main parts of the manuscript.

Minor revision  

  1. Although this study mentions that all subjects undergo multimodal imaging, the image shown here are only color fundus images. Figures should be re constituted displayed by multimodal image data for each patients. Using photoshop is strongly recommended.

As suggested by the reviewer structural OCT images were added to those previously provided.

  1. Detailed explanation of ATN grading is required in method section.

A new table was added with a detailed explanation of the ATN grading system to go with the existing citations.

  1. P value of Table 2 seems to have less meaning. Authors should consider incorporating Table 3 into Table2.

    We thank the reviewer for the suggestion. We proceeded to combine both tables and modified accordingly through the manuscript

Reviewer 3 Report

Comments and Suggestions for Authors

The Ruiz‑Medrano et al reported the correlation of posterior staphyloma types  and the myopic maculopathy. This is a well-written manuscript and the design of this study is sound. Also, the clinical relevance of this study is high. Still, there are some question concering the formatting.

1. The abstract should be merged into one solitary section without the bald subheading (baclground, method, etc).

2. Too many sections in the introduction, should be merged into 3-4 sections.

3. The method section line 76-101 should be merged into one section.

4. The discussion section should be merged into 6-7 sections. 

If completed these modification, the manuscript can be accepted.

Author Response

The Ruiz‑Medrano et al reported the correlation of posterior staphyloma types  and the myopic maculopathy. This is a well-written manuscript and the design of this study is sound. Also, the clinical relevance of this study is high. Still, there are some question concering the formatting.

  1. The abstract should be merged into one solitary section without the bald subheading (baclground, method, etc).
  2. Too many sections in the introduction, should be merged into 3-4 sections.
  3. The method section line 76-101 should be merged into one section.
  4. The discussion section should be merged into 6-7 sections. 

If completed these modification, the manuscript can be accepted.

Thank you very much for your suggestions. We proceeded to apply all of them, reducing the number of sections of the introduction, methods and discussion sections and reformatting the abstract.